# Hypoxia Engineered Bone Marrow Mesenchymal Stem Cells Targeting System with Tumor Microenvironment Regulation for Enhanced Chemotherapy of Breast Cancer

**DOI:** 10.3390/biomedicines9050575

**Published:** 2021-05-19

**Authors:** Jingzhi Zu, Liwei Tan, Li Yang, Qi Wang, Jing Qin, Jing Peng, Hezhong Jiang, Rui Tan, Jian Gu

**Affiliations:** 1College of Medicine, Southwest Jiaotong University, Chengdu 610031, China; zjz578060490@163.com (J.Z.); rafael0927@163.com (L.T.); 2College of Life Science and Engineering, Southwest Jiaotong University, Chengdu 610031, China; wangqi20210122@163.com (Q.W.); QJ1473829841@163.com (J.Q.); pj554820@163.com (J.P.); jianghz10@sina.com (H.J.); 3Sichuan Purity Pharmaceutical Co., Ltd., 3-1 Jiuxing Ave, Hi-Tech Zone, Chengdu 610041, China; 4School of Materials Science and Engineering, Southwest Jiaotong University, Chengdu 610031, China; l.yang2012@foxmail.com; 5School of Pharmacy, Southwest University for Nationalities, Chengdu 610051, China

**Keywords:** DTX@H-bMSCs, chemotherapy, tumor targeting, inflammatory regulation, combination therapy

## Abstract

Improving the tumor targeting of docetaxel (DTX) would not only be favored for the chemotherapeutic efficacy, but also reduce its side effects. However, the regulation of the tumor microenvironment could further inhibit the growth of tumors. In this study, we introduced a system consisting of hypoxia-engineered bone marrow mesenchymal stem cells (H-bMSCs) and DTX micelles (DTX-M) for breast cancer treatment. First, the stem cell chemotherapy complex system (DTX@H-bMSCs) with tumor-targeting ability was constructed according to the uptake of DTX-M by hypoxia-induced bMSCs (H-bMSCs). DTX micellization improved the uptake efficiency of DTX by H-bMSCs, which equipped DTX@H-bMSCs with satisfactory drug loading and stability. Furthermore, the migration of DTX@H-bMSCs revealed that it could effectively target the tumor site and facilitate the drug transport between cells. Moreover, in vitro and in vivo pharmacodynamics of DTX@H-bMSCs exhibited a superior antitumor effect, which could promote the apoptosis of 4T1 cells and upregulate the expression of inflammatory factors at the tumor site. In brief, DTX@H-bMSCs enhanced the chemotherapeutic effect in breast cancer treatment.

## 1. Introduction

Docetaxel (DTX) as the first-line chemotherapy drug has been widely used in cancer therapy [1,2]. However, although DTX has a significant antitumor effect, its side effects cause great pain in cancer patients [3]. Surprisingly, with the development of nanotechnology, the targeting drug delivery system has demonstrated some success in certain cancer types of clinical setting and could increase the accumulation of the drug at the tumor site [4]. Moreover, based on special tumor microenvironment (TME), such as low pH, glutathione (GSH) overexpression, hypoxia, and inflammation, the controlled release behavior of DTX was realized by a series of intelligent response materials [5]. These excellent achievements not only increase the therapeutic effect, but also decrease the side effects of chemotherapy [6]. However, the biocompatibilities of the exogenous materials have been evaluated, and unexpected opsonization by reticuloendothelial system (RES) could also result in the clearance of most drugs during systemic circulation [7]. The metabolism and biosafety of drug carriers are still the main challenge for nanomedicine [8], furthermore, due to the limitation of dose threshold, passive tumor targeting based on the enhanced permeability and retention (EPR) effect cannot significantly improve the tumor growth inhibition by nanomedicines [9]. Moreover, since cancer is a complex disease, targeted enhancement and multitarget synergistic therapy have become the research trend of current treatment programs [10]. Therefore, natural cells with hypo-immunogenicity, inherent tumor-tropic, and migratory properties have attracted an increasing amount of attention as drug delivery systems [11,12].

Over the past few years, stem cells have not only achieved great success in regenerative medicine but have also shown huge potential for drug delivery [13,14]. Recent studies have suggested that in some treatments, such as cancer [15], central nervous system injury [16], or cardiovascular diseases [17], stem cells could deliver drugs [18] to the lesions, and could also act as the carrier of oncolytic virus [19] and tumor-related apoptosis-inducing factors [20] to tumor sites, and further promote the therapeutic effect by self-secretion [21,22]. Therefore, a stem cells targeting system would be a candidate for cancer therapy. In addition, the homing ability of stem cells is based on a special signal from the lesions. Accordingly, upregulating the expression of the homing-relative chemokine receptor could improve the homing ability of stem cells [23]. Results from TME research have indicated that hypoxia stimulation may improve the migration of stem cells to tumor through the C-X-C chemokine receptor type 4 (CXCR4)/stromal cell-derived factor-1α (SDF-1α) axis [24,25]. Therefore, it would be feasible to enhance the homing effect of stem cells by hypoxic pretreatment. Conversely, several challenges are presented by these treatments, such as drug-loading, stability, and cytotoxicity of anticancer drugs to stem cells [26]. Fortunately, as we have described earlier, nanotechnology could significantly improve cellular efficiency, and maintain the chemical structure and activity of the drug molecule [27,28]. Therefore, incorporating nanoparticles instead of free chemotherapeutic drugs into the stem cells could avoid direct exposure, slow down the release behavior during systemic circulation, and improve the transferred dose of the drugs.

In addition, in cancer therapy, stem cells could be used as functional cells for TME regulation in the inhibition of tumor growth [29]. Recent developments have indicated that stem cells play an important role in immune balance regulation [30]. As an important immune mechanism, inflammation regulated tumor growth, metastasis and apoptosis [31]. Inflammation is one of the special phenomena in the TME. Therefore, the regulation of inflammatory factor expression in the TME might favor cancer therapy [32]. Consequently, targeted delivery of DTX nanoparticles by hypoxia-induced stem cells would not only provide an effective way to improve chemotherapy efficiency and reduce side effects, but also elucidate the regulation of the TME, which has not been investigated in detail. Moreover, the combination therapy with DTX for tumors could be realized.

In this study, as shown in Scheme 1, bone mesenchymal stem cells (bMSCs) are a type of tissue stem cell with cross embryonic differentiation potential and high self-replication, which are easily isolated and have low immunogenicity [30,33]. Therefore, we first upregulated CXCR4 expression in bMSCs to increase homing ability to tumors by hypoxia induction. DTX was encapsulated into the amphipathic polymer poly (ethylene oxide) monomethyl ether-poly(lactide-*co*-glycolide) (mPEG-PLGA) to prepare DTX micelles (DTX-M). Then, DTX@H-bMSCs were constructed by the incorporation of DTX-M and hypoxia-induced bMSCs (H-bMSCs). The drug loading, drug-transfer process, and pharmacodynamics of DTX@H-bMSCs were investigated in vitro. Moreover, we explored the effect of DTX@H-bMSCs on inflammatory regulation in the TME. Finally, the homing ability and the antitumor effect of DTX@H-bMSCs were evaluated in a subcutaneous tumor model of breast cancer in mice, and the relevant pathological indices were used to further investigate the mechanism.

## 2. Materials and Methods

### 2.1. Materials

Docetaxel was supplied by Sichuan Xieli Pharmaceutical Co., Ltd. (Sichuan Xieli Pharmaceutical, Chengdu, China). The block copolymer mPEG-PLGA (Da: 2k-2k) was provided by Xi’an ruixi Biological Technology Co., Ltd. (Xi’an ruixi Biological Technology, Xi’an, China). 4′,6-Diamidino-2-phenylindole (DAPI), 5-Bromo-2-deoxyUridine (BrdU), and Coumarin-6 (C6) were obtained from Sigma-Aldrich (Sigma-Aldrich, St.Saint Louis, MO, USA). Cyanine 7 NHS ester near-infrared fluorescent dye (Cy7) was supplied by Biotium (Biotium, Hayward, CA, USA). Acetonitrile and acetic acid were purchased from Sigma-Aldrich (Sigma-Aldrich, Darmstadt, Germany). Annexin V-FITC/PI apoptosis detection kit and Calcein/PI cell viability/cytotoxicity assay kit were purchased from Jiangsu KeyGEN BioTECH Co., Ltd. (Jiangsu KeyGEN BioTECH, Jiangsu, China).

For Western blot and immunofluorescent staining assay, antibodies against CXCR-4, GAPDH, and horseradish peroxidase (HRP) conjugated secondary antibodies were purchased from Wuhan Servicebio Technology Co. Ltd. (Wuhan Servicebio Technology, Wuhan, China). SDF-1α, TNF-α, BrdU mouse McAb antibody and Ki67 rabbit monoclonal antibody were obtained from Affinity Biosciences (Affinity Biosciences, Cincinnati, OH, USA).

4T1 tumor cell line was available from the American Type Culture Collection (ATCC, Rockville, MD, USA) and grown in Roswell Park Memorial culture media (RPMI 1640, Gibco, CA, USA) supplemented with 10% fetal bovine serum (FBS, Gibco, CA, USA) and 1% penicillin-streptomycin (RPMI 1640, Gibco, CA, USA). The cell culture was maintained in a 37 °C incubator with a humidified 5% CO_2_ atmosphere.

Five-week-old female BALB/c mice used for antitumor tests were purchased from Dashuo Bio-Technology Co., Ltd. (Dashuo Bio-Technology, Chengdu, China) throughout the experiment, and were housed at a temperature of 20 ± 2 °C relative humidity of 50–60%, and with 12 h light–dark cycles. bMSCs were extracted from the bone marrow of tibia and femur of 4–6 weeks old male BALB/c mice. The protocol was authorized by the Institutional Animal Care and Use Committee of Chengdu Military General Hospital.

### 2.2. Preparation and Characterization of DTX-M

The mPEG–PLGA was purchased from Xi’an ruixi Biological Technology Co., Ltd. (Sichuan Xieli Pharmaceutical, Chengdu, China), and which was synthesized by ring-open polymerization of ε-lactic and ε-glycolic acid on mPEG by using Sn(Oct)2 as a catalyst. DTX-M was prepared by thin-film hydration method as described previously [34]. First, 10 mg of DTX and 190 mg mPEG-PLGA were codissolved in 4 mL dehydrated alcohol in a round-bottomed flask. The solvent was evaporated in rotator evaporator at 60 °C with mixing, and a thin layer of homogenous film. The mixture was hydrated with normal saline at 60 °C under moderate shaking. In this condition, the amphiphilic copolymers self-assembled into micelles with DTX encapsulated in, and the suspension was filtered through a 0.22 μm syringe filter (Millipore Co., Darmstadt, Germany). Freeze-dried micelles were reconstructed with normal saline, and the DTX micelles were redissolved under mild stirring at an appropriate temperature. The experiments were performed according to previous research.

The surface charge distributions, and particle size distributions were determined by dynamic light scattering (DLS) (Malvern Panalytical, London, UK), and the morphology was observed by transmission electron microscopy (TEM) (JEOL, Tokyo, Japan). X-ray diffraction (XRD) (Empyrean, Almelo, Netherlands) spectra of DTX powder, DTX-M, and the physical mixture of DTX and copolymers were obtained using Bruker D2 Phaser Diffractometer (Bruker, Tübingen, Germany). The sampling parameter was the step range of 5–50° at a speed of 2°/min.

### 2.3. Drug Loading and Encapsulation Efficiency of DTX-M

The drug loading (DL) and entrapment efficiency (EE) of the micelles were determined using HPLC (Agilent, Palo Alto, CA, USA) with a C18 column (4.6 × 150 mm × 5 µm, Grace Analusis Column). The composition of the mobile phase was acetonitrile/ammonium (45/55, *v/v*) at a flow rate of 1 mL/min. Detection was recorded on a diode array detector (1260 DAD VL) at a wavelength of 232 nm. Before measurement, the samples were diluted with the mobile phase [4]. The results were calculated using the following equations:

DL% = amount of DTX determined in micelle/(amount of DTX determined + copolymer) × 100%

EE% = amount of DTX determined in micelle/amount of DTX in feed × 100%


### 2.4. The In Vitro Release Behavior of DTX-M

The in vitro release kinetics of DTX from micelles or free DTX were performed using a dialysis method [35]. In this study, 0.5 mL of DTX micelles or free DTX solution with 500 μg DTX was placed in a dialysis bag (molecular mass cut-off 8 kDa). The dialysis bags were incubated in 40 mL of phosphate buffer (pH = 7.4) containing 0.5% polysorbate 80 at 37 °C with gentle shaking (100 rpm). Polysorbate 80 in the release media is used as the solubilizer to improve the solubility of DTX in media, facilitating the release of DTX into dissolution medium from the dialysis bags. At predetermined intervals, 2 mL aliquots were withdrawn, and the dissolution medium was replaced with prewarmed fresh medium. The released DTX was quantified using the HPLC method as described previously, and the cumulative release profile over time was demonstrated.

### 2.5. Isolation and Identification of bMSCs

The bMSCs were extracted from the bone marrow cavity of 4-6 weeks old female BALB/c mice as previously reported. The bMSCs were cultured in low-glucose Dulbecco’s Modified Eagle’s Medium (DMEM) containing 10% FBS, 1% penicillin and streptomycin at 37 °C in an incubator with 5% CO_2_. Cells from the third to sixth passages were used for all experiments. The expression of cellular surface markers CD 29, CD 45, and CD 90 was determined by flow cytometry BD Biosciences, Franklin Lakes, NJ, USA) to identify the type of isolated bMSCs [36,37].

### 2.6. Hypoxia Induction of bMSCs

The H-bMSCs were engineered by culturing bMSCs under hypoxic conditions for 12, 24, 48 or 72 h, the oxygen volume fraction is 2 %. The expression of CXCR4 on H-bMSCs was used as an index for the screening of the induction process. CXCR4 was detected on the surface of H-bMSCs using immunofluorescence and Western blot.

### 2.7. Construction and Characterization of DTX@H-bMSCs

Since DTX@H-bMSCs were constructed by the uptake of DTX-M by H-bMSCs, the hydrophobic fluorescent probe Coumarin-6 (C6) was used to replace DTX and encapsulated into the micelles for the intuitive optimization of the preparation process. The H-bMSCs were incubated in 6-well plates (3 × 10^4^ cells/well) for 24 h at 37 °C, and the free C6 and C6-M (concentration of C6 was 200 ng/well) were added to each plate. After incubation for different times (1, 2, 4, and 6 h), the samples were harvested for observation under laser confocal fluorescence microscopy (Nikon, Tokyo, Japan), and fluorescence intensity was detected using flow cytometry. Moreover, the optimized process was used to construct DTX@H-bMSCs, and the DL of per cell was detected using HPLC as follows: the samples were collected after 4 h incubation, DTX feeding from 10 to 40 μM, and 1.5 mL anhydrous methanol was added for resuspension. The supernatant was obtained after cell disruption for 5 min (8 w), ultrasonic treatment for 15 min, vortex for 10 min, and centrifugation for 10 min (1000 rpm). The drug contents in the supernatant were detected by HPLC, and the chromatographic conditions are described in the previous section.

### 2.8. Cytotoxicity of DTX Micelles for bMSCs

The cytotoxicity of DTX micelles for bMSCs and H-bMSCs was evaluated using MTT assay. First, the cells (3 × 10^3^ cells/well) were cultured in 96-well plates and incubated for 24 h at 37 °C. Micelles of different concentrations were added, and cultivation continued for 24 h. MTT solution (5 mg/mL, 20 µL) was added to each well. Thereafter, the MTT solution was removed and replaced with DMSO (160 μL/well) after 3 h. The absorbance was measured at a wavelength of 490 nm using a 680-model microplate reader from Infinite M200 PRO Multimode Microplate Reader (Tecan Trading, Männedorf, Switzerland).

### 2.9. The In Vitro Release Behavior of DTX@H-bMSCs

C6 was evaluated instead of DTX for in vitro release behavior and tested in transwell plates with a 0.4 μm pore membrane (Corning 24 mm Transwell^®^) that would only allow the passage of released micelles or drug molecules. The C6@H-bMSCs (4 × 10^4^ cells/cm^2^) were planted in the upper chamber, and the samples were harvested at the set time point. The results were analyzed using flow cytometry. In addition, we planted the 4T1 cells (1 × 10^4^ cells/cm^2^) in the lower chamber to uptake C6 or C6-M, which would be harvested at the set time and detected using flow cytometry. The release behavior of C6@H-bMSCs and the uptake of 4T1 reflect the drug transfer rate between cells.

### 2.10. Drug Delivery from DTX@H-bMSCs to 4T1 Cells

Similar to the previous section, C6 was chosen, instead of DTX, for further study as a fluorescent probe. The C6@H-bMSCs (2 × 10^4^ cells/well) and the BrdU-labeled 4T1 cells (2 × 10^4^ cells/ well) were cocultured in 6-well plates, allowing the C6-M or C6 that was released from the C6@H-bMSCs to be taken up by 4T1 cells. At the set time point, fluorescence images of the cells were observed by laser confocal fluorescence microscopy.

### 2.11. Inhibition of DTX@H-bMSCs on Tumor Growth In Vitro

First, 4T1 cells transfected with luciferase (Luc-4T1) were used to evaluate the antitumor effect of DTX@H-bMSCs. The DTX@H-bMSCs and Luc-4T1 were cocultured in 6-well plates, and the total number of cells was 4 × 10^4^. The setting range of DTX@H-bMSCs:Luc-4T1 was from 1:5 to 1:1. After 12, 24 and 48 h incubation, the luciferin solution (150 μg/mL, 100 μL) was added, and luciferase expression was measured by the 680-model microplate reader to further calculate the survival rate of Luc-4T1 cells in each group.

### 2.12. Live/Dead Cell Staining Assay

The DTX-M (3 μg/mL), bMSCs (2 × 10^4^ cells/well), H-bMSCs (2 × 10^4^ cells/well), and DTX@H- bMSCs (2 × 10^4^ cells/well) were placed in the upper chamber with a 8 μm pore membrane (Corning 6.5 mm Transwell^®^), and DTX in each group measured 3 μg. Then, the 4T1 cells (2 × 10^4^ cells/well) were seeded in the lower chamber. After incubation for 24 h, cells in the lower chamber were washed thrice with PBS, 200 μL working solution (10 mL containing 2 μM Calcein AM, 8 μM PI) was added, and the cells were further incubated for 30 min at 25 °C away from light. Finally, the cells were washed thrice with PBS, and observed under fluorescence microscope.

### 2.13. Evaluation of Apoptosis

Similar to the live/dead cell staining assay, the therapeutic systems were placed in the upper chamber of the transwell plates, and the 4T1 cells (2 × 10^4^ cells/well) were seeded in the lower chamber. Apoptosis was investigated with an Annexin V-FITC Apoptosis Detection kit according to the manufacturer’s protocol. After 24 h incubation, the cells of different groups were treated accordingly, and trypsinised, and harvested by centrifugation at 200× *g* at 4 °C for 15 min. The cell aggregate was suspended in 0.1 mL Annexin-V binding buffer and then incubated with FITC-conjugated Annexin-V for 15 min at room temperature, avoiding light. Propidium iodide (PI, 1 µg/mL) was added to the samples immediately prior to flow cytometry analysis. The stained samples were analyzed using flow cytometry, and the percentage of apoptotic cells was determined by the Annexin V/PI ratio. Enzyme linked immunosorbent assay was used to determine the concentration of TNF-α in the medium of each group.

### 2.14. In Vitro Migration Assay

Migration of the different engineered bMSCs to the targets were conducted using 8.0 μm pore membranes (Corning 6.5 mm Transwell^®^). bMSCs or H-bMSCs (1×10^4^ cells/well) were seeded in the upper chamber, and NS, 4T1 cells, or SDF-1α medium were placed into the lower chamber, respectively. After 24 h incubation, the cells on the lower surface of the membrane were fixed with methanol and stained with 0.1% crystal violet for 30 min. Cells that had migrated were observed and counted in three randomly selected microscopic fields (400×) using an Olympus microscope (Olympus, Tokyo, Japan).

### 2.15. Biodistribution Evaluation of DTX@H-bMSCs In Vivo

In this section, the near-infrared fluorescent dye Cy7 was used as the fluorescent probe instead of DTX. 4T1 cells (1 × 10^6^ cells/0.1 mL) were incubated in the right flank of the athymic female BALB/c mice by subcutaneous injection. The bMSCs or H-bMSCs were coincubated with Cy7-M (0.1mL, 1 mg/mL) for 4 h before injection. The mice were randomly divided into Control, Cy7-M, Cy7@bMSCs, and Cy7@H-bMSCs group until the tumor diameter was approximately 8 mm. Each mouse was injected with 1×10^6^ Cy7@H-bMSCs or Cy7@bMSCs, and Cy7 in each group measured 100 μg. The mice were anesthetized with 1% pentobarbital sodium (50 mg/kg,) before near-infrared imaging at predetermined time points. Real-time monitoring was detected using a fluorescence imaging system (PerkinElmer, Waltham, MA, USA; excitation = 740 nm, emission = 790 nm long pass) at the time intervals of 4, 12, 24, 48, and 72 h. At the 24 h post-injection, the mice were sacrificed, and the heart, liver, spleen, lung, kidney, and tumor were harvested to evaluate the fluorescence intensity of the ex vivo organs. The quantitative fluorescence intensity was acquired using onboard software.

### 2.16. BrdU for the In Vivo Trace of bMSCs

BrdU (10 μg/mL) was incubated with bMSCs, H-bMSCs for fluorescence labeling. Cells (1 × 10^5^) were injected into tumor-bearing mice. At 24 and 72 h of post-injection, the tumor tissues were collected and the BrdU fluorescence images were observed under a fluorescence microscope. Moreover, the CXCR4/SDF-1α axis represents mechanism, migration of H-bMSCs, and the expression of SDF-1α in tumors was detected by immunofluorescence staining. Tumor tissues were prepared into sections with a thickness of 7 μm and fixed with tetramethyl benzidine. The sections were then incubated with the corresponding primary and secondary antibodies and observed under fluorescence microscope.

### 2.17. Antitumor Effect of DTX@H-bMSCs In Vivo

The antitumor activity of DTX@H-bMSCs was evaluated in a 4T1 cell breast orthotopic implantation tumor model. The mice were injected with cell suspension (100 μL) containing 1 × 10^5^ 4T1 cells per mouse. When the tumor volume approached 100 mm^3^, the mice were randomly divided into control, DTX-M, DTX@bMSCs and DTX@H-bMSCs group in a double-blinded manner, which were intravenously injected with NS (Control), DTX-M (dose of DTX, 3 mg/kg), DTX@bMSCs (dose of DTX, 3 mg/kg) and DTX@H-bMSCs (dose of DTX, 3 mg/kg) every four days. Tumor size and body weight were measured every other day during the experimental period and the tumor volume was calculated using the formula: volume (mm^3^) = length × width^2^ × 0.5. The mice were sacrificed when the tumor diameter reached 20 mm, and the tissues were excised for further analysis. In addition, the survival time of the mice was observed, and the mice were considered dead when tumor volume reached 4000 mm^3^.

### 2.18. Histopathological Test

Tumor and normal tissues were fixed and post-fixed in 4% paraformaldehyde for at least 24 h. Then, the samples were embedded in paraffin and coronally sectioned at approximately 7 μm for a series of detection experiments. Sections were stained with hematoxylin and eosin (HE). The histomorphology of tumor tissues and normal tissues was observed under a microscope. To further investigate the therapeutic effect, apoptosis of tumor cells was detected using terminal deoxynucleotidyl transferase-mediated nick-end labeling (TUNEL) (Promega, Madison, WI, USA) staining assay, and tumor cell proliferation was analyzed by immunohistochemical Ki 67 staining (Lab Vision & NEOMARKERS, Waltham, MA, USA). Tumor tissues were fixed in 15%wt formaldehyde once the mice were sacrificed. The TUNEL method was performed according to the manufacturer’s instructions, and the Ki 67 staining was processed using the labeled streptavidin-biotin method. The apoptosis and proliferation of tumor cells were detected using the fluorescence microscope (Olympus, Tokyo, Japan). In addition, since inflammation is an important mechanism for tumor growth, we investigated the expression of inflammatory factors in each group. The immunofluorescence antibodies were specially connected to the target protein, and the methods provided by the manufacturer were followed. Fluorescence was observed under a fluorescence microscope.

### 2.19. Western Blot for CXCR4 Expression Characterization

The expression levels of CXCR4 were identified by Western blot. Western blot analyses were performed according to the protocols for the routine with antibodies against CXCR4.

### 2.20. RNA Extraction and qPCR

The miRNA transcriptional expression of CXCR4 in H-bMSCs was investigated using qPCR. Total mRNA was extracted from cells using TRIZOL extraction. Both the amount and purity of the RNA were confirmed by measuring the absorbance ratio at 260/280 nm. Total RNA (1 µg) was converted to cDNA using a PrimeScript TM RT reagent kit with gDNA Eraser and PCR amplification followed by an ABI Step One Plus instrument and software (Applied Biosystems, Foster City, CA, USA) using SYBR Green PCR Master Mix. The RNA levels of the target genes were normalized by GAPDH according to the ∆∆Ct method. Each procedure was independently performed in triplicate to ensure minimal bias. The primers used in this study were as follows:


*CXCR4:5′-CGTGAATGAGTGTCTAGGCAGG-3′,*



*5′-GGCTTTGGTTTTAAGTGCCATC-3′;*



*GAPDH:5′-GGCACAGTCAAGGCTGAGAATG-3′,*



*5′ATGGTGGTGAAGACGCCAGTA-3′;*


### 2.21. Immunofluorescence Staining

Immunofluorescence staining was used to observe the expression of relative proteins. For in vitro evaluation, the cells were rinsed with PBS and fixed with 4 % paraformaldehyde for 15 min, followed by permeabilization for 30 min. After blocking with 3 % BSA for 1 h, the cells were incubated with a primary antibody against GAPDH or CXCR4 at 4 °C overnight. After rinsing, the cells were incubated with a secondary antibody for 2 h at room temperature. The nuclei were stained with DAPI (0.5 μg/mL, Vector Laboratories, Burlingame, CA, USA) for 5 min. For the in vivo experiments, the tissues were prepared as 7 μm sections, washed twice in PBS, permeabilized (in PBS with 0.1% Triton X-100), and blocked (in PBS containing 1 % BSA and 0.3 % Triton X100) for 1 h at room temperature. The sections were then incubated with the specific primary antibodies, as follows: rabbit anti-SDF-1α, TNF-α, for 3 d at 4 °C. After being rinsed with PBS, the sections were incubated with appropriate secondary antibodies overnight at 4 °C. After rinsing with PBS, sections were mounted on slides with Vectashield (Vector Laboratories, Inc., Burlingame, CA, USA). Immunofluorescence confocal microscopy was performed using confocal laser scanning microscope.

### 2.22. Statistical Analysis

The comparison of each group was evaluated by statistical analysis using SPSS software with one-way analysis of variance (ANOVA). All statistical results are expressed as the mean ± standard deviation (SD), with an alpha level of 0.05.

## 3. Results

### 3.1. Preparation and Characterization of DTX-M

To our knowledge, nano-formulations could improve cellular uptake efficiency. Considering its biocompatibility, the amphiphilic block copolymer mPEG-PLGA has been shown to be a favorable carrier of micelles, which could improve the dispersion of hydrophobic drugs in water and prolong the internal circulation time [38,39]. Furthermore, in our previous study, mPEG-PLGA was used as a carrier to deliver the natural active drug luteolin for ischemic stroke therapy, which improved the neuroprotective effect [34]. Thus, in this study, mPEG-PLGA was chosen to form the DTX micelles and exhibited superior stability and drug loading capacity. DTX micelles were prepared by the thin film hydration method. As shown in Figure 1A, DTX-M was clear and transparent, with a light blue milky solution. Compared with water, the obvious Tyndall effect appeared in DTX-M. The mean particle size of DTX-M and the black micelles was around 30 nm (Figure 1B). The zeta potential of DTX-M was approximately 0 mV (Appendix A). The morphology of DTX-M was observed under TEM, which was shown a homogeneous spheroid, and the size was consistent with measurements (Figure 1B). Furthermore, crystallographic analyses were performed using XRD, and the result is shown in Figure 1C. In comparison with blank mPEG-PLGA and DTX, the graph of DTX-M lacked the characteristic diffraction peaks, indicating that DTX was completely encapsulated in the micelles. The DL and EE of DTX-M with different drug feeding were detected using HPLC. The properties of DTX-M as displayed in Table 1, indicated increased drug feeding from 2% to 5%, while DL increased from 1.91± 0.032% to 4.82 ± 0.174%, EE maintained above 95%, and particle size measured approximately 30 nm with a narrow PDI value, indicating that the morphology was relatively homogeneous. A further increase of drug feeding to 12% was observed and DL increased to 8.27 ± 0.925%, while EE decreased to only 68.92 ± 7.71%. The results suggested the use of DTX-M with 5% DL for further exploration.

Moreover, the in vitro release behavior of the micelles is shown in Figure 1D. In the free DTX group, 94.9% DTX was released into the media within 24 h, and the micelle group released approximately 18.4%. With the prolongation of release time, 63% DTX was released from the micelles after 72 h. This suggested that the micelles could not only prevent burst release, but also sustain drug release. In addition, due to the special microenvironment of tumor tissues, we explored the stability of DTX-M under normal physiological conditions (pH 7.4) and in a weak acidic environment (pH 6.5). As shown in Figure 1E,F, the particle size in both pH 7.4 media and pH 6.5 media were maintained at about 30 nm during the first 12 h. However, when the storage time longer than 24 h, the particle size in pH 6.5 increased dramatically while that in pH 7.4 media remained stable. It is generally known that the pH value of the cytoplasm is similar to the physiological conditions (approximately pH 7.4), thus, DTX-M could retain stability in bMSCs. Furthermore, the instability of micelles in pH 6.5 indicated that the DTX micelle responded to micro-acid environment, which induced the release of DTX from the micelles. These results illustrated that DTX-M was stable in bMSCs, but accelerated drug release at the tumor site due to its slightly acidic TME. In conclusion, DTX-M was successfully produced.

### 3.2. DTX@H-bMSCs Construction

The bMSCs were isolated from the tibial bone marrow cavity of BALB/c mice (4-6 weeks old), and cell type was verified by detection of cell surface antigens. As shown in Appendix A, special antigens, CD90 and CD29 were highly expressed in about 90% of the cells, whereas CD45 was not. This result indicated that bMSCs were isolated from the tibial bone marrow cavity. Next, we investigated the effect of hypoxia condition on regulating CXCR4 expression in bMSCs. As shown in Figure 2A, the transcriptional expression of CXCR4 increased remarkably after 6 h of hypoxia. The fold change of CXCR4 miRNA transcriptional expression during 24 and 48 h hypoxia was 5.4 and 5.8 times higher than that of the control, respectively. Furthermore, the protein expression of CXCR4 could be upregulated by hypoxia condition, this was particularly observed with the prolongation of hypoxia condition at 48 h, which was more significant. In addition, CXCR4 expression at 72 h hypoxia was not significantly higher than that at 48 h (Figure 2B,C). Western blot and immunofluorescence staining of CXCR4 exhibited a consistent phenomenon (Figure 2D). While hypoxic condition lasted for 48 h, the viability of bMSCs decreased to 60% in Figure 2E. Therefore, hypoxia induction for 24 h would be a feasible method to prepare engineered bMSCs overexpressing CXCR4.

Furthermore, the DTX@H-bMSCs complex system was constructed via the cellular uptake of DTX-M. First, the cytotoxicity of DTX-M for bMSCs and H-bMSCs was evaluated. As shown in Figure 3A,B, the cell viability showed the same tendency in the bMSCs and H-bMSCs group. In comparison with free DTX, the cytotoxicity of DTX-M decreased significantly. The cell viability of the free group decreased with an increase in the dosage of DTX. Moreover, the cell viability of the free group was only 40%, while that of the DTX-M group was still about 80% at DTX concentration of 40 μM. In addition, with further increases in the dosage, the cell viability decreased sharply. It can be concluded that the cytotoxicity of DTX to bMSCs and H-bMSCs was lower than that to 4T1, which may be due to the nonproliferative fibroblastic state adopted by bMSCs [40]. Thus, these results suggested that 40 μM might be the highest concentration for DTX-M to construct the DTX@bMSCs and DTX@H-bMSCs complexes. To observe the uptake of DTX and DTX-M by bMSCs and H-bMSCs directly, C6 was used as a fluorescence indicator to replace DTX to be loaded into the micelles. As shown in Figure 3C, the fluorescence of the C6-M group was brighter than that of the free C6 group at each time point. To quantify the cellular uptake of the C6, the intensity of bMSCs and H-bMSCs treated with different formulations, was detected using flow cytometer, and the results are shown in Figure 3D and Appendix A, respectively, which is consistent with the fluorescence images. Moreover, the fluorescence intensity of bMSCs and H-bMSCs remarkably increased within 4 h, and there is no significant difference between them. Upon incubation for a further 6 h, the fluorescence intensity does not increase significantly. Therefore, these results showed that DTX-M was more suitable than free DTX, and 4 h coincubation would be used for the construction of DTX@H-bMSCs. Furthermore, we detected the drug-loading of DTX@H-bMSCs using HPLC. The results are displayed in Figure 3E and Appendix A. With drug feeding from 10 to 40 μM, the DL of DTX@H-bMSCs increased from 10.01 ± 2.15 to 145.91 ± 5.54 pg/cell, and the DTX@bMSCs increased from 11.11 ± 0.35 to 152.90 ± 0.47 pg/cell. In addition, with the further increase in drug feeding, the DL of the complexes increased but did not change significantly. The intracellular DTX with 40 μM drug feeding was also measured after incubation for different times (0.5, 1, 2, 4, 6, and 8 h), as shown in Appendix A, it could be found the same tendency between C6 (Figure 3D) and DTX. Considering the cytotoxicity of DTX, the 4 h coincubation time and 40 μM drug feeding would be the optimal process for DTX@H-bMSCs construction. In brief, the DTX@H-bMSCs complex system with highly efficient cargo loading properties was constructed successfully.

### 3.3. Drug Transferred In Vitro

The drug transfer efficiency between cells is an important ability of this cellular drug delivery system. First, the drug release behavior and drug delivery of DTX@H-bMSCs in vitro were evaluated using flow cytometry. In this study, C6 was replaced with DTX for entrapment in the micelles. As shown in Figure 4A and Appendix A, about 48% and 42% C6 were released from C6@bMSCs and C6@H-bMSCs within 2 h respectively, while 72 h incubation was required for more than 90% C6 release. This suggested that DTX would be released continuously from both C6@bMSCs and C6@H-bMSCs within 72 h. Next, to investigate drug delivery in vitro, the C6@bMSCs and C6@H-bMSCs were independently displayed in the upper chamber of the transwell plate, and 4T1 cells were cultured in the lower chamber. As shown in Figure 4B, the fluorescence intensity of 4T1 cells increased significantly within 12 h, and then slowly. Similar to the results of in vitro release, the trend of fluorescence intensity of 4T1 cells was consistent with that of C6@bMSCs and C6@H-bMSCs. Therefore, the drugs could be released from the complex and taken up by target cells. We further incubated the C6@H-bMSCs with the BrdU-labeled 4T1 cells, and the fluorescence distribution in cells was observed using laser confocal microscopy. As shown in Figure 4C, both green and red were observed in the same cells, which indicated that C6 could be transferred between the cells. In addition, the green fluorescence density in the C6@H-bMSCs group was greater than that in the C6@bMSCs group, which might be attributed to the change in migration of the engineered bMSCs.

As the classical mechanism of homing, the expression regulation of the CXCR4/SDF-1α transfer axis is considered a major mechanism of migration for H-bMSCs. To demonstrate this conjecture, transwell migration assays were used for evaluation. In Figure 5A,B, compared with bMSCs, the number of H-bMSCs in the SDF-1α (+) group was higher than that in the bMSCs group. Furthermore, it could be easily found that H-bMSCs had a stronger tendency to 4T1 cells than bMSCs. These results suggested that CXCR4 overexpression could enhance the migration of H-bMSCs to 4T1 cells. In brief, the engineered bMSCs exposed to hypoxia could effectively transfer the drug to the tumor cells based on the CXCR4/SDF-1α axis.

### 3.4. Inhibition of Tumor Growth In Vitro

First, Luc expression from the Luc-4T1 cells was used to evaluate the inhibition of DTX@H-bMSCs to tumor cells in vitro. As shown in Figure 6A, cell viability of 4T1 cells decreased with increased bMSCs or H-bMSCs. It was revealed that bMSCs and H-bMSCs inhibited the growth of 4T1 cells in vitro and enhanced the chemotherapeutic effect of DTX. Moreover, the inhibitory effect of DTX@H-bMSCs was superior to that of DTX@bMSCs, which might be attributed to their affinity for 4T1 cells. Furthermore, as shown in Figure 6B, the live/dead cell image demonstrated the effect of DTX@H-bMSCs on 4T1 cells. Compared to the control group, the number of dead 4T1 cells increased significantly (from 12% to 45%) in the bMSCs, H-bMSCs, DTX-M, DTX@bMSCs, and DTX@H-bMSCs group. Moreover, the number of dead cells in each group was as follows: DTX@H-bMSCs group > DTX-M group > DTX@bMSCs group > H-bMSCs > bMSCs. These results suggested that DTX and bMSCs have a synergistic inhibitory effect on the growth of 4T1 cells. In addition, the migration of the H-bMSCs to 4T1 cells based on the CXCR4/SDF-1α axis might be the reason that the antitumor effect of DTX@H-bMSCs was superior to that of DTX@bMSCs. Flow cytometry was used to evaluate apoptosis of 4T1 cells in each group. As shown in Figure 6D, the apoptosis rate in the treatment groups was higher than that in the control group. Similar to the live and dead cell assays, the apoptosis rate was as follows: DTX@H-bMSCs group > DTX@bMSCs group > DTX-M group > H-bMSCs > bMSCs, and the rate of late apoptosis was higher than that of early apoptosis. In addition, few necrotic cells could be found in each group, which suggested that the cytotoxicity of both single DTX and DTX@H-bMSCs on 4T1 cells was due to apoptosis rather than necrosis. The flow cytometry results were consistent with the MTT and live/dead cell assays, suggesting that the DTX@H-bMSCs group could effectively suppress the proliferation of 4T1 cells in vitro.

Surprisingly, the bMSCs or H-bMSCs could not only inhibit the growth of 4T1 cells in vitro, but also exert synergistic effect with DTX-M. Research has indicated that regulation of inflammatory factor expression is an important mechanism for stem cells in disease treatment. Inflammation expression in the TME is higher than that in normal tissue, which regulates tumor growth and metastasis [31]. Moreover, the overexpression of inflammatory factors could induce tumor immune response and apoptosis. Therefore, the inhibitory effect of bMSCs or H-bMSCs on 4T1 cells might be due to the regulation of inflammatory factor expression in the TME. In this study, the inflammatory factor TNF-α was used as a detection index. As shown in Figure 6C, after coculture for 24 h, the concentrations of TNF-α in the bMSCs and H-bMSCs group were significantly higher than that in the control and DTX-M groups, demonstrating the up-regulatory effect of bMSCs and H-bMSCs on inflammatory factor expression in the TME. Consequently, we concluded that DTX@H-bMSCs or DTX@bMSCs inhibited 4T1 cell growth via regulation of inflammatory expression in the TEM.

### 3.5. Biodistribution of DTX@H-bMSC In Vivo

The in vivo targeting ability of DTX@H-bMSCs for 4T1 cell tumor engraftment was investigated in BALB/c mice. The subcutaneous tumor model mice were randomly divided into four groups: control, Cy7 micelles (Cy7-M), Cy7@bMSCs, and Cy7@H-bMSCs. The Cy7 fluorescence chromogenic agent replaced DTX for exploration in vivo, and the results are displayed in Figure 7A–C. At 4 h post-injection, using the same dose of Cy7, fluorescence signals could be detected in experimental groups, and the fluorescence began to concentrate at the tumor sites. As incubation time increased, tumor site accumulation of the fluorescent signals in the Cy7-M group reached a peak level at 12 h, which was stronger than that in the bMSCs group. After 12 h, the fluorescence intensity in the Cy7-M group weakened, whereas its intensity increased in the bMSCs group at the tumor site. These results indicated that micelles could be concentrated at the tumor site according to the EPR effect [41], but clearance during systemic circulation was higher than that of the bMSCs carriers. Furthermore, the fluorescence intensity of the Cy7@H-bMSCs group was significantly higher than that of the Cy7@bMSCs group, which suggests that the CXCR4/SDF-1α axis could improve migration of bMSCs to tumor tissues. The fluorescence intensity in the Cy7@bMSCs and Cy7@H-bMSCs group decreased gradually at 48 and 72 h post-injection and could not be detected in the Cy7-M group. This phenomenon could be attributed to the fact that the non-antigenic bMSCs could prevent rapid clearance of the drug to maintain extended circulation in vivo. In addition, the peak fluorescence value in the bMSCs group was observed 24 h post-administration. Therefore, to further observe the biodistribution in major organs, the mice were sacrificed 24 h post-injection, and the fluorescence signals in ex vivo tissues are displayed in Figure 7B–D. Consistent with the live images, fluorescence intensity at the tumor site was observed as follows: Cy7@H-bMSCs group > Cy7@bMSCs group > Cy7-M group > control. Fluorescence intensity was stronger in tumor compared with in normal tissues. Further, the fluorescence intensity of tumors in the Cy7@H-bMSCs group was significantly higher than that of normal tissues, which indicates superior tumor homing ability.

Moreover, bMSCs and H-bMSCs labeled with BrdU were injected into subcutaneous tumor model mice to study their precise concentration at tumor sites at 24 and 72 h. At the same time, SDF-1α at the tumor site was labeled with red fluorescence. As displayed in Figure 7E, we found that SDF-1α was widely expressed at tumor sites, and both H-bMSCs and bMSCs could be observed. Moreover, the fluorescence intensity of BrdU after three days was higher than that after one day. This might be due to the proliferation of bMSCs and H-bMSCs at the tumor site. Furthermore, compared with the bMSCs group, the fluorescence intensity of BrdU in H-bMSCs group was much stronger, suggesting that hypoxia induction could promote accumulation of bMSCs at the tumor site. Importantly, it was found that the distribution of H-bMSCs at the tumor site was consistent with that of SDF-1α. Currently, the promotion of bMSCs migration to tumor is mainly achieved through the CXCR4/SDF-1α axis, which are thought to be the most important pair of cytokines [25]. These results revealed that the CXCR4/SDF-1α axis is not only related to the tumor-homing ability of H-bMSCs, but also affected its distribution at the tumor site. Consequently, H-bMSCs could achieve tumor-targeted delivery and retention of drugs.

### 3.6. Therapeutic Effect In Vivo

Encouraged by the favorable therapeutic effect in vitro and the satisfactory accumulation at the tumor site, the antitumor effect of DTX@H-bMSCs was further explored in vivo. The mouse model of breast cancer in situ construction and the administration route are shown in Figure 8A. The tumor growth curves and the image of mice at real-time points showed the same trend (Figure 8B,C). Compared with the DTX-M group, DTX@bMSCs and DTX@H-bMSCs showed better inhibition of tumor growth, with an inhibitory rate of approximately 65% and 80%, respectively. Moreover, the antitumor effect of DTX@H-bMSCs is superior to that of DTX@bMSCs, which might be due to improved tumor homing ability. Furthermore, the isolated tumors were harvested 16 days post-inoculation, and the volume and weight of the tumors in each group showed that DTX and bMSCs exerted a synergistic effect on tumor growth inhibition, which would be further improved by the hypoxia engineering of bMSCs (Figure 8D). We further observed the survival time of tumor-bearing mice, and the results are shown in Figure 8E. Mice in the control group treated with NS began to die at 16 days post-administration, and the remaining mice died within 32 days. Compared with the control group, the treatment group could extend the survival time of 4T1 tumor bearing mice. In addition, 20% of mice in the DTX-M group, 50% in DTX@bMSCs, and 80% in DTX@H-bMSCs survived 36 days after administration. These results suggested that bMSCs combined with chemotherapy could effectively improve the survival rate and time.

Moreover, the mechanisms of the antitumor effect were investigated by histopathological analysis. First, the HE staining was used to detect tumor cell damage (Figure 9A). From the eosin distribution in the tumor slides, a large number of dead cells could be observed in the DTX@H-bMSCs group. Compared with the DTX-M group, the dead cells in the DTX@H-bMSCs group were more concentrated in the depth of the tumor tissue. Next, Ki 67 and TUNEL staining were used to evaluate proliferation and apoptosis in each group, respectively. Similar to the results of HE staining, the greatest number of apoptotic cells were observed in the DTX@H-bMSCs group (Figure 9B). Surprisingly, more TUNEL positive cells were observed in the interior of the tissue rather than the surface in the DTX@H-bMSCs group compared with the DTX-M group (Figure 9C). This may be due to overexpression of SDF-1α in tumor tissue, which mediated the migration of DTX@H-bMSCs into the interior for chemotherapy.

Inflammatory infiltration is an important factor in tumorigenesis and apoptosis [42], and inflammation in the TME can initiate the tumor immune response [43,44]. Fortunately, bMSCs have been shown to regulate inflammation in previous in vitro investigations. Therefore, we further observed the distribution of the inflammatory factors in the tumor in each group. As shown in Figure 10, a large amount of TNF-α was detected in the control group, and the deeper the tumor, the higher the density of TNF-α. Compared to the DTX-M group, inflammation in the bMSCs group was more obvious, which is consistent with in vitro experimentation. Therefore, the antitumor effect of the bMSCs may be due to the upregulation of inflammatory factors in the TME. Surprisingly, when we observed the tumor tissue of each group, the expression of TNF-α in the DTX@H-bMSCs group was significantly higher than that in other groups, which suggests that hypoxia engineering of bMSCs improved the upregulation of inflammatory expression in the TME. In addition, the phenomenon of the DTX@H-bMSCs group from the previous pathological images might also be due to the homing of the CXCR4 receptor on the surface of H-bMSCs to SDF-1α in the dense inflammatory area in the tumor. In brief, these results illustrated that the DTX@H-bMSCs targeting system with TME regulation could enhance the chemotherapeutic effect on breast cancer in vivo.

### 3.7. Safety and Biocompatibility Evaluation

Surprisingly, the weight of mice in the DTX@bMSCs and DTX@H-bMSCs group remained normal throughout the treatment period (Figure 11A) and did not appear to produce side effects. The biocompatibility was evaluated by HE staining of major tissues (Figure 11B), which indicates that the DTX@H-bMSCs did not cause any histological damage to the heart, liver, spleen, lung, and kidney of mice. These results suggested that the DTX@H-bMSCs did not show obvious toxicity or side effects during tumor treatment.

## 4. Discussion and Prospect

MSCs with low immunogenicity, inherent tumor eosinophilic and migration characteristics have become a research hotspot in regenerative medicine and as drug delivery [33,45]. Moreover, the studies of tumor microenviroment have shown that the homing ability of stem cells is established on the basis of special signals of lesions, such as TNF-α, TGF-β and other cytokines secreted in tumor microenvironment [46], as well as soluble factors secreted by tumors [47]. Currently, the promotion of stem cells migration to tumor is mainly achieved through the CXCR4/ SDF-1α axis, and the expression of CXCR4 in stem cells can be further enhanced by hypoxia induction [48,49]. Therefore, there is potential to upregulate the expression of CXCR4 for promoting the migration of stem cells to tumor sites. In addition, the homing effect of MSCs is believed to occur in a chemokine-oriented manner under the condition of continuous inflammation [50,51], and the proinflammatory cytokines in tumors can upregulate the adhesion proteins required for the chemotactic movement of MSCs [52]. Therefore, regulation of the expression of inflammatory cytokines in TME may be beneficial to cancer treatment [32]. However, there are still many technical bottlenecks to be overcome. Such as, the large-scale and standardized preparation process of the stability stem cells need to be further developed [45]. For the tumor therapy, the mechanism of the stem cells was still controversial [53]. Therefore, there is still a long way to go before stem cell therapies would be converted to large scale clinical trials. For another part of the synergistic system, DTX micelles have shown great potential in market, such as the Genexol-PM [54] in South Korea. Therefore, we could have a prediction that if the industrialized technical bottlenecks of stem cells could be overcome, the clinical transformation of DTX@H-bMSC would be realized.

## 5. Conclusions

In summary, we constructed and characterized a novel targeted chemotherapy system based on hypoxia-engineered bMSCs. The pharmacodynamics of this system have been investigated in detail using mouse models of breast cancer. In this study, the H-bMSCs exhibited tumor homing ability based on the CXCR4/SDF-1α axis and the effect of inflammatory regulation in the TME. Moreover, as a drug delivery system, H-bMSCs could efficiently and accurately transfer DTX nano-medicine to the interior of tumor tissues, which improved the enrichment and retention of DTX and bMSCs. Importantly, DTX@H-bMSCs effectively prolonged the survival time of breast cancer model mice, owing to its superior tumor inhibition effect and low side effects. Overall, this synergistic strategy for pairing a chemotherapeutic drug with engineered bMSCs exhibited potential for clinical transformation and application, presenting strong implications and future research directions for breast cancer treatment.

## Data Availability

Not applicable.

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
