# Peer review of "Hypoxia Engineered Bone Marrow Mesenchymal Stem Cells Targeting System with Tumor Microenvironment Regulation for Enhanced Chemotherapy of Breast Cancer"

_biomedicines, 2021, doi:10.3390/biomedicines9050575_

Round 1
Reviewer 1 Report
Breast cancer is a serious health problem. PLGA nanoparticles can be used succesfully in breast cancer therapy, without sidde effect. Congratulations for your hard work.

Author Response
Responses to the Reviewer 1 Comments:
Thank you very much for your kindness and patience to review our article entitled as “Hypoxia Engineered Bone Marrow Mesenchymal Stem Cells Targeting System with Tumor Microenvironment Regulation for Enhanced Chemotherapy of Breast Cancer”.
We have revised the whole manuscript according to your advice. The language has been further polished. And the corrections could be found as the "Track Changes" in Microsoft Word format in the revised manuscript. The followings are the responses to the reviewer 1 comments.
Point 1: in the last part of the introduction - line 78 appears a part of the chapter of materials and methods.
Response 1: Thank you very much. We did a shrot summarize in the last part of the introduction section to introduce what we would be studied, which included the objection, materials, methods and predicted results. Therefore, the line 78 to line 90 could be thought of as a summary of schemes.
Point 2: Scheme 1 should be moved to materials and methods.
Response 2: Thanks for your advice. The Scheme 1 was the content of graph which summarized the methods, the results and the mechanism. Therefore, Scheme 1 was arranged as the last paragraph of the introduction section.
Point 3: what other similar studies have been performed so far according to the literature?
Response 3: Thanks very much for your advice. Over the recent years, MSCs have attracted more and more attention as drug delivery system by its low immunogenicity, inherent tumor eosinophilic and migration characteristics. Moreover, MSCs not only exhibited the excellent homing ability to the target, but also could be used as an active agent for treatment. Therefore, the MSCs targeting system development is a wide range of research hot spot, especially in cancer therapy. For examples, the enhancement of anti-tumor activity is achieved in MSCs loaded with paclitaxel-PLGA nanoparticles for glioma-targeting therapy [1]. MSCs loaded with paclitaxel in-PLGA nanoparticles show a remarkably different targeting capability from that of nanoparticles in mouse orthotopic lung tumor model [2]. Another in vivo test indicate that MSCs with different oHSV variants (MSCs-oHSV) can significantly prolong the survival of brain tumor-bearing mice with intracarotid administration [3]. And it is also reported that MSCs have been used to deliver tumor necrosis factor related apoptosis-inducing ligand, and can home to tumor sites in a lung metastatic cancer model [4]. In a conclusion, MSCs targeting system would be a candidate for cancer therapy. The related discussions have been added in the revision.
Point 4: why PEG-PLGA was used and how they were synthesized; a brief description of these particles would be required.
Response 4: Thanks for your advice. The amphiphilic block copolymers mPEG-PLGA with its biocompatibility has been proved to be favor to the carrier of micelles, which could improve the dispersion of hydrophobic drugs in water and prolong the internal circulation time [5, 6]. Not only that, in our previous study, the mPEG-PLGA was used as the carrier to deliver luteolin (a natural active drug) for ischemic stroke treatment, which has been published in Pharmaceutics [7]. Therefore, in the present study, the mPEG-PLGA was also chosen to form the DTX micelles, exhibiting superior stability and drug loading capacity as expected.
In addition, the mPEG–PLGA in this study was purchased from Xi'an ruixi Biological Technology Co., Ltd, and which was synthesized by ring-open polymerization of ε- lactic and ε-glycolic acid on mPEG by using Sn(Oct)2 as catalyst. And the synthetic products was purified by dialysis. Finally, as discribed in the methods section, the DTX micelles were prepared by thin-film hydration method, and the characterization of physicochemical property were investigated by drug loading capacity, morphology and drug release behavior. The related discription and discussions have been added in the revision.
Point 5: at the line 391, appears a strange sign (in chinese, I think !!)
Response 5: Thank you very much, we are terribly sorry for this stupid mistake. We have made the correction and carefully checked all the revision.
Point 6: in the chapter on results and discussions, only the results and their interpretation are presented. a small comparison with the literature, would be useful.
Response 6: Thanks very much for your suggestion, the further relative discussions have been added in the revised manuscript.
Point 7: It is a difficult article to follow, maybe separating the results in vivo and in vitro would be much more user to follow
Response 7: Thanks for your advice. After the construction of this collaborative treatment system (DTX@bMSCs and DTX@H-bMSCs), the in vitro evaluations were firstly carried out, including drug delivery behavior (Figure 4), the migration and the pharmacodynamic (Figure 6). During the in vitro study, the therapeutic mechanism and efficacy could be achieved. However, the ex vivo data generally could not be fully representative of efficacy in vivo. Therefore, we conducted the targeting (Figure 7), therapeutic (Figures 8-10) and safety evaluation (Figures 11) of the optimized formulations in vivo, and analyzed the results. The logic of this paper has been strengthened in the revision.
Thank you again for your valuable comments and suggestions.
References:
- Wang, Xiaoling.; Gao, Jianqing.; Ouyang, Xumei.; Junbo; Sun, Xiaoyi.; Yuanyuan. Mesenchymal Stem Cells Loaded with Paclitaxel-poly(lactic-co-glycolic acid) Nanoparticles for Glioma-targeting Therapy. Int. J. Nanomed. 2018, 13, 5231-5248.
- Sadhukha, Tanmoy.; O'Brien, Timothy D.; Prabha, Swayam. Nano-engineered Mesenchymal Stem Cells as Targeted Therapeutic Carriers. J. Control. Release 2014, 196, 243-251.
- Du, W.; Seah, I.; Bougazzoul, O.; Choi, G.H.; Meeth, K.; Bosenberg, M.W.; Wakimoto, H.; Fisher, D.; Shah, K. Stem Cell-Released Oncolytic Herpes Simplex Virus has Therapeutic Efficacy in Brain Metastatic Melanomas. P. Natl. Acad. Sci. USA. 2017, 114, e6157-e6165.
- Loebinger, M.R.; Eddaoudi, A.; Davies, D.; Janes, S.M. Mesenchymal Stem Cell Delivery of TRAIL Can Eliminate Metastatic Cancer. Cancer Res. 2009, 69, 4134-4142.
- Photos, P.J.; Bacakova, L.; Discher, B.; Bates, F.S.; Discher, D.E. Polymer Vesicles In Vivo: Correlations with PEG Molecular Weight. J. Control. Release 2003, 90, 323-334.
- Vasir, J.K.; Labhasetwar, V. Biodegradable Nanoparticles for Cytosolic Delivery of Therapeutics. Adv. Drug Deliv. Rev. 2007, 59, 718-728.
- Tan, L.; Chen, L.; Wang, Y.; Yu, J.; Zeng, S. Pharmacodynamic Effect of Luteolin Micelles on Alleviating Cerebral Ischemia Reperfusion Injury. Pharmaceutics 2018, 10, 248.

Reviewer 2 Report
The manuscript entitled “Hypoxia Engineered Bone Marrow Mesenchymal Stem Cells Targeting System with Tumor Microenvironment Regulation for Enhanced Chemotherapy of Breast Cancer” is focused on the development of a complex system (DTX micelles uptaken by hypoxia-engineered bone marrow mesenchymal stem cells) designed for improving DTX efficacy in breast cancer. This is an interesting research with a huge number of experiments suitable for biomedicines journal. However, there are several points that should be addressed before its publication.
- DTX is a highly lipophilic drug. Its encapsulation makes easy its administration. This reinforces the use of drug delivery systems.
- Line 38-40. Nanotechnology helps to get a selective location of the drug at tumor sites. However, this is not recent. Nanoparticles (liposomes, micelles, albumin nanoparticles) loaded with PTX and DOX have ben approved since early nineties by FDA or EMA, so its use in cancer its not new. I suggest changing the sentence.
- It is true that the use of nanomedicines is challenged. However, there are numerous approved formulations in cancer disease and other pathologies (e.g. infectious diseases). I suggest changing lines 47-48. As written, it is seemed that nanomedicines are complex and not used in clinic and that is not true.
- What about the clinical translation of these systems? they are complex.
- Sentence 84-85 should be reviewed.
- Scheme 1 is a little blurry. I suggest increasing image quality.
- I suggest checking table 1. What did the authors mean with the numbers in the first row? 1., 2…..Moreover, what did the author mean with 9…..14. in the second and third row? I guess that all these numbers are a mistake and should be deleted.
- Lines 356-357. It is true that to obtain PDI values lower than 0.2 its is difficult. However, PDI values in the range of 0.1-0.4 (data from DLS analysis) indicate a moderate polydispersion that is acceptable, but it is not a monodispersed distribution. This sentence should be corrected.
- Line 370. Why did the authors indicate that around 94% of free DTX was released? Released from the dialysis bag? Did they mean dissolved? If you do not have a DDS you do not have release profile. I guess that the authors incorporated a DTX solution inside this bag.
- As indicated in the figure 1, DTX release is slow. After 72 hours just 60% of DTX was released. Did the authors evaluate more time points? A 60% release is poor. Specially considering that during 48 h just the 20% of DTX is released?
- I was wondering, did the authors evaluated the DTX simulating the conditions of stem cells? At pH 6.5 micelles are not stable (as indicated by size changes) after 24 h of incubation. This could affect DTX release.
- Please, check line 391. There is a typo.
- Did the authors evaluate the release of coumarin from the micelles?
- At 40McM DTX has a considerable antiproliferative effect against stem cells, so it is not the most appropriate concentration to confirm that. Probably, the use of lower concentration (20 or perhaps 30 McM) is better, as you are using stem cells as DTX carriers.
- In efficacy studies: did the authors evaluated the effect of non-loaded stem cells?
Author Response
Responses to the Reviewer 2 Comments:
Thank you very much for your kindness and patience to review our article entitled as “Hypoxia Engineered Bone Marrow Mesenchymal Stem Cells Targeting System with Tumor Microenvironment Regulation for Enhanced Chemotherapy of Breast Cancer”.
We have revised the whole manuscript according to your advice. The language has been further polished. And the corrections could be found as the "Track Changes" in Microsoft Word format in the revised manuscript. The followings are the responses to the reviewer 2 comments.
Point 1: DTX is a highly lipophilic drug. Its encapsulation makes easy its administration. This reinforces the use of drug delivery systems.
Response 1: Thank you for sure. In recent decades, DTX micelles have been extensively studied to improve their dispersion. The combination of multi-modal treatment can further improve therapeutic effect of cancer, because it is a complex and multi-target disease. Therefore, we chose the engineered bone mesenchymal stem cells as drug carrier to deliver the DTX micelles to tumor sites, which could further improve targeting effect of DTX. At the same time, based on the regulation of stem cells in tumor microenvironment, the combination therapy with DTX for tumors could be explored. The DTX@H-bMSCs system could be a good candidate for the cancer therapy.
Point 2: Line 38-40. Nanotechnology helps to get a selective location of the drug at tumor sites. However, this is not recent. Nanoparticles (liposomes, micelles, albumin nanoparticles) loaded with PTX and DOX have been approved since early nineties by FDA or EMA, so its use in cancer its not new. I suggest changing the sentence.
Response 2: Thanks very much for your suggestions. We have updated the review about the developments of nanomedicines in revised manuscript.
Point 3: It is true that the use of nanomedicines is challenged. However, there are numerous approved formulations in cancer disease and other pathologies (e.g. infectious diseases). I suggest changing lines 47-48. As written, it is seemed that nanomedicines are complex and not used in clinic and that is not true.
Response 3: Thanks very much for your advice. We feel so sorry for our less rigorous. Indeed, nanomedicines, such as Genexol, Nanoxel-PM, Abraxane and Apealea, have been used in clinic for the cancer therapy. We have made the modification in the revision.
Point 4: What about the clinical translation of these systems? they are complex.
Response 4: Thank you very much. Stem cell therapy has been widely used to treat a variety of diseases, exhibiting a huge potential in clinical translation. However, there are still many technical bottlenecks to be overcome. Such as, the large-scale and standardized preparation process of the stability stem cells need to be further developed [1]. Moreover, the quality standards of stem cell have not been formulated, so it is necessary to strictly supervise the effectiveness and safety of stem cell technology. For cancer therapy, the mechanism of the stem cells is still controversial [2]. So, there is still a long way to go before stem cell therapy could be converted to large scale clinical trials. For another part of the synergistic system, DTX micelles has shown great potential in market, such as the Genexol-PM in South Korea [3]. So, we could have a prediction that if the industrialized technical bottlenecks of stem cell therapy could be overcome, the clinical transformation of DTX@H-bMSCs would be improved.
Point 5: Sentence 84-85 should be reviewed.
Response 5: Thank you very much. The sentence 84-85 has been corrected in revised manuscript.
Point 6: Scheme 1 is a little blurry. I suggest increasing image quality.
Response 6: Thank you very much. The high-quality images have been added in revised manuscript.
Point 7: I suggest checking table 1. What did the authors mean with the numbers in the first row? 1., 2…..Moreover, what did the author mean with 9…..14. in the second and third row? I guess that all these numbers are a mistake and should be deleted.
Response 7: Thank you very much. There might be an error in the format after uploading the files, and the numbers in the Table 1 have been corrected.
Point 8: Lines 356-357. It is true that to obtain PDI values lower than 0.2 its is difficult. However, PDI values in the range of 0.1-0.4 (data from DLS analysis) indicate a moderate polydispersion that is acceptable, but it is not a monodispersed distribution. This sentence should be corrected.
Response 8: Thanks very much for your advice. What we want to express was that the morphology of micelles was relatively homogeneous, and the size was all around 30 nm. The description about PDI have been modified in revised manuscript.
Point 9: Line 370. Why did the authors indicate that around 94% of free DTX was released? Released from the dialysis bag? Did they mean dissolved? If you do not have a DDS you do not have release profile. I guess that the authors incorporated a DTX solution inside this bag.
Response 9: Thank you very much. In the in vitro release behavior of DTX-M, 0.5 mL of free DTX solution with 500 μg DTX was placed in a dialysis bag, which were incubated in phosphate buffer (pH=7.4) containing 0.5 % polysorbate 80, then the DTX was released into dissolution medium from the dialysis bags. In this study, polysorbate 80 in the release media was used as the solubilizer to improve the solubility of DTX in media. Moreover, the molecular mass cut-off of the dialysis bag was 8 kDa, hence the free DTX could dissolve in outside of dialysis bag. And samples were obtained at the setting time and detected by HPLC for the content of DTX.
Point 10: As indicated in the figure 1, DTX release is slow. After 72 hours just 60% of DTX was released. Did the authors evaluate more time points? A 60% release is poor. Specially considering that during 48 h just the 20% of DTX is released?
Response 10: Thanks for your advice. The in vitro release behavior results showed that the drug could be released from the micelles. Compared with free DTX, it could be found that the micelles delayed the release of the drug, indicating that the micelles could protect the structure of the drug, and also reduced the cytotoxicity of DTX, which remained a relatively stable state in bMSCs. Then the release rate of free DTX in 72 h was almost 100 %, so we only showed 72 h in Figure 1D, which proved that the micelles could improve the stability of DTX and achieve the effect of sustained slow release for a long time.
Point 11: I was wondering, did the authors evaluated the DTX simulating the conditions of stem cells? At pH 6.5 micelles are not stable (as indicated by size changes) after 24 h of incubation. This could affect DTX.
Response 11: Thank you very much. It could be observed from Figure 1E and 1F during the first 12 h, the particle size in both pH 7.4 media and pH 6.5 media maintained about 30 nm. However, when the storage time longer than 24 h, the particle size in pH 6.5 increased dramatically while that in pH 7.4 media kept stable. It is known that the pH value of the cytoplasm is similar to the physiological conditions (pH 7.4), so that the DTX-M could keep stability in bMSCs. Furthermore, the unstable of micelles in pH 6.5 indicated that the DTX micelles had the response to micro-acid environment, which induced the release of DTX from the micelles. These results indicated that DTX-M was stable in the bMSCs, but accelerated drug release at the tumor sites due to its slightly acidic tumor microenviroment. The discussions have been supplemented in revision.
Point 12: Please, check line 391. There is a typo.
Response 12: Thank you very much for your suggestions. We are very sorry for this stupid mistake. The revised manuscript has been corrected.
Point 13: Did the authors evaluate the release of coumarin from the micelles?
Response 13: Thanks for your constructive suggestions. In this study, the hydrophobic fluorescent probe Coumarin-6 (C6) instead of DTX was encapsulated into the micelles for the observation and investigation of the micellar intercellular delivery. Since either DTX or C6 was respectively encapsulated in the micelle form by the same copolymer mPEG-PLGA, the DTX micelles and C6 micelles should be consistent in the cell uptake and in vitro release behavior. These properties had been verified in DTX@H-bMSCs construction part. In addition, it could be found that the same tendency between C6 and DTX from the Figure 3D, Figure S1 and Table S2. Besides, C6 has been widely used to replace DTX, PTX and other chemotherapy drugs for detecting the in vitro delivery of nanocarriers [4]. And C6 also is wrapped in PCL-NPs to study the delivery behavior of marrow-isolated adult multilineage inducible (MIAMI) cells in malignant glioma [5]. Therefore, The available data supports the existing conclusion, and the further discussions have been added in the revised manuscript.
Point 14: At 40McM DTX has a considerable antiproliferative effect against stem cells, so it is not the most appropriate concentration to confirm that. Probably, the use of lower concentration (20 or perhaps 30 McM) is better, as you are using stem cells as DTX carriers.
Response 14: Thanks very much for your suggestions. Firstly, at 40 μM DTX-M,the cell viability of bMSCs and H-bMSCs was 67.45 % and 76.12 %, while at 20 μM DTX-M, the cell viability of bMSCs and H-bMSCs was 71.41 % and 70.08 %. And there was no significant difference between the two concentrations. And at 40 μM free DTX,the the cell viability of bMSCs and H-bMSCs decreased to 39.63 % and 47.30 %. Secondly, in the Table S1 of Supporting Information (SI), the intracellular DTX of DTX@bMSCs and DTX@H-bMSCs at 40 μM was almost twice as much at 30 μM. Therefore, in consideration of cytotoxicity and intracellular DTX, the concentration of 40 μM DTX was used for investigation in detail.
Point 15: In efficacy studies: did the authors evaluated the effect of non-loaded stem cells?
Response 15: Thanks very much for your suggestions. The inhibitory effect of non drug-loaded stem cells and DTX micelles on tumor cells was compared in vitro (Figure 6), and found that the anti-tumor effect of DTX micelles was significantly better than that of non drug-loaded stem cells. In addition, the mechanism of stem cells in tumor therapy is still controversial [2], some views posit that MSCs primarily via exosomes, polarization of phagocytic monocytes, and other paracrine effects achieve therapeutic effect [6, 7]. However, the rapid proliferation, differentiation and immunosuppressive action of stem cells are beneficial to tumor growth [8]. Therefore, the tumor inhibition of non drug-loaded stem cells in vivo had not been studied, but mainly focused on the enhancement of tumor microenvironment regulation on tumor chemotherapy.
Finally, thank you for all the important comments and suggestions, which are valuable in improving the quality of our manuscript. Thank you again for your contributions and consideration.
References:
- Krueger, Timothy E. G.; Thorek, Daniel L. J.; Denmeade, Samuel R.; Isaacs, John T.; Brennen, W. Nathaniel. Concise Review: Mesenchymal Stem Cell-Based Drug Delivery: The Good, the Bad, the Ugly, and the Promise. Stem Cell. Transl. Med. 2018, 7, 651-663.
- Lazennec, G.; Jorgensen, C. Concise Review: Adult Multipotent Stromal Cells and Cancer: Risk or Benefit? Stem Cells 2010, 26, 1387-1394.
- Kim, T.Y.; Kim, D.W.; Chung, J.Y.; Sang, G.S.; Bang, Y.J.; Kim T.Y.; Kim D.W.; Chung J.Y.; Shin S.G.; Kim S.C.; Heo D.S.; Kim N.K.; Bang Y.J. Phase I and Pharmacokinetic Study of Genexol-PM, a Cremophor-free, Polymeric Micelle-Formulated Paclitaxel, in Patients with Advanced Malignancies. Clin. Cancer Res. 2004, 10, 3708-3716.
- Sadhukha, Tanmoy.; O'Brien, Timothy D.; Prabha, Swayam. Nano-engineered Mesenchymal Stem Cells as Targeted Therapeutic Carriers. J. Control. Release 2014, 196, 243-251.
- Mathilde Roger.; Anne Clavreul.; Marie-Claire Venier-Julienne.; Catherine Passirani.; Laurence Sindji.; Paul Schiller.; Claudia Montero-Menei.; Philippe Menei. Mesenchymal Stem Cells as Cellular Vehicles for Delivery of Nanoparticles to Brain Tumors. Biomaterials 2010, 31, 8393-8401.
- Phinney, Donald G.; Pittenger, Mark F. Concise Review: MSC-Derived Exosomes for Cell-Free Therapy. Stem Cells 2017, 35, 851-858.
- Prockop, Darwin J.; Youn Oh, Joo. Mesenchymal Stem/Stromal Cells (MSCs): Role as Guardians of Inflammation. Mol. Ther. 2012, 20, 14-20.
- Yu, Ji Min.; Jun, Eun Sook.; Bae, Yong Chan.; Jung, Jin Sup. Mesenchymal Stem Cells Derived From Human Adipose Tissues Favor Tumor Cell Growth in Vivo. Stem Cells & Dev. 2008, 17, 463-473.

Round 2
Reviewer 2 Report
The authors have improved the quality of the manuscript by adressing all the comments and suggestions. Consequently, it deserves to be published in the current (revised) form